# Transcutaneous Vagus Nerve Stimulation for Insomnia in People Living in Places or Cities with High Altitudes: A Randomized Controlled Trial

**DOI:** 10.3390/brainsci13070985

**Published:** 2023-06-22

**Authors:** Liang Zhang, Yinchuan Jin, Qintao Zhang, Hongyao Liu, Chen Chen, Lei Song, Xiao Li, Zhujing Ma, Qun Yang

**Affiliations:** Department of Military Medical Psychology, Air Force Military Medical University, Xi’an 710032, China; o1342635018@outlook.com (L.Z.); jyc_fmmu@163.com (Y.J.); scda_liuhongyao@163.com (H.L.); a1342635018@outlook.com (L.S.); lX_fmmu@outlook.com (X.L.); mzj_fmmu@163.com (Z.M.)

**Keywords:** transcutaneous vagus nerve stimulation, insomnia, plateau, cognitive behavioral therapy for insomnia

## Abstract

Background: The purpose of this study was to investigate the effectiveness and safety of transcutaneous vagus nerve stimulation (tVNS) to improve insomnia in the special environment of a plateau. Methods: This study was a single-center, single-blind, randomized controlled trial. A total of 100 patients with insomnia at high altitude were randomized into three groups receiving either transcutaneous vagus nerve stimulation intervention in the left ear tragus (treatment group), pseudo-stimulation intervention (sham group), or cognitive behavioral therapy for insomnia (CBTI group). The primary measure was the Pittsburgh Sleep Quality Index (PSQI) score. In addition, we assessed the patients’ objective sleep status with polysomnography and evaluated changes in the Insomnia Severity Index Scale (ISI) and Generalized Anxiety Disorder-7 (GAD-7) scores. We used one-way ANOVA and repeated-measures ANOVA for analysis. Results: Patients’ PSQI, ISI, and GAD-7 scale scores significantly decreased after 4 weeks of tVNS treatment and were greater than those of the control group. Polysomnographic data also demonstrated shortened sleep latency and longer deep sleep in the patients. Conclusion: tVNS is effective in improving sleep quality and reducing anxiety levels in high-altitude insomnia patients but should be confirmed in future adequate and prolonged trials to guide clinical promotion.

## 1. Introduction

Insomnia at high altitudes is an independent condition commonly found in plateau areas, and it is also a risk factor for acute and chronic plateau diseases [1], mainly manifested as difficulty in sleep initiation or maintenance, or even sleepless nights, accompanied by symptoms such as fatigue, depression, panic and chest tightness, and decreased brain cognitive function. The environmental factors such as low pressure and lack of oxygen are the main reasons for insomnia and sleep structure changes in a plateau [2]. Under the unique natural environmental conditions of a plateau, the body’s neural and respiratory regulation function and circadian physiological rhythm are altered, and unique periodic breathing alternately manifests at night as repeated deep and fast breathing and apnea, resulting in frequent nocturnal apnea and awakening, leading to changes in sleep structure, an increase in light sleep, a decrease in deep sleep, an increase in the total number and duration of awakenings, obstructed breathing during sleep, frequent awakenings, and a decrease in sleep quality, resulting in insomnia [3,4,5]. Sleep plays a vital role in an individual’s ability to learn, their physical activity, and their work performance. Chronic sleep loss can negatively affect both physical and mental health, increasing the risk of developing several physiopsychic diseases.

Currently, psychotherapy and medication are the two major forms of treatment for insomnia at high altitudes. Cognitive behavioral therapy for insomnia (CBTI) has been recommended by the American Academy of Sleep Medicine and the Chinese Sleep Research Society as the preferred treatment option for chronic insomnia disorder [6,7,8], but psychotherapy has limited efficacy for episodes of insomnia, is expensive, and has a long treatment period, making it difficult to be popularized. Pharmacological treatment can alleviate sleep problems in a short period, but side effects such as tolerance and excessive addiction induced by drugs cannot be ignored [9,10,11]. In these circumstances, there is an urgent need for a drug-free alternative intervention for the effective and safe treatment of insomnia in plateaus.

Sleep is influenced by both individual perceptions and emotions, as well as by the dual regulation of nerves and fluids, of which the vagus nerve, due to its wide distribution and action, has a regulatory role in sleep and mood. Numerous studies have used the peripheral channel that regulates activity in the brainstem, thalamus, cerebral cortex, and other related areas, the vagal branch of the vagus nerve, for therapeutic purposes [12,13,14]. Due to the non-invasive nature of transcutaneous vagus nerve stimulation, there is increasing interest in its use in basic, translational, and clinical research. Transcutaneous vagus nerve stimulation has recently been demonstrated in several studies to help treat primary insomnia and enhance sleep in healthy subjects [15,16], indicating that the stimulation of the vagus nerve may support the maintenance of a flexible balance in the autonomic nervous system, which is required to promote wakefulness and sleep states.

In addition, several studies have shown that the stimulation of the vagus nerve can affect the secretion of neurotransmitters in sleep-related brain regions, altering the excitability of the cerebral cortex and reducing the production of inflammatory factors through the cholinergic anti-inflammatory pathway [17,18,19]. Meanwhile, patients with altitude insomnia often have an increase in neurogenic inflammatory factors [20] and an imbalance in neurotransmitter regulatory mechanisms, which provides a rationale for using tVNS to intervene in high-altitude sleep problems. On this premise, we conducted a 2-month, single-blind, randomized controlled trial to evaluate whether tVNS can improve insomnia symptoms caused by environmental factors at high altitudes and provide a new and convenient method to improve sleep problems in people living at high altitudes.

## 2. Materials and Methods

### 2.1. Study Design

This single-blind, randomized, controlled trial investigated the effects of a 4-week tVNS intervention on subjective and objective sleep indicators in people with plateau insomnia. At the trial site, which was in Lhasa, Tibet, at a height of 3650 m, 105 patients signed up for the study. A team of clinicians with the appropriate training, including neurologists and psychiatrists, handled the patient recruitment procedure. The recruited patients were randomly divided into three groups: the tVNS group, the control group (sham tVNS), and the CBTI group. Patients in the tVNS group received tVNS stimulation at 25 Hz in the tragus of the left ear for five consecutive interventions per week for a total of 20 interventions over four weeks. Patients in the control group received only 30 s of tVNS stimulation in the tragus region of the left ear and then stopped stimulation, and the other settings were kept the same as those in the tVNS group. Patients in the CBTI group received psychotherapy including sleep restriction, stimulation control, breathing relaxation, sleep hygiene education, and cognitive therapy at different time points for a total of four weeks. Patients were assessed at the following time points: baseline, end of treatment (4 weeks after the start of treatment), and follow-up (4 weeks and 8 weeks after the end of treatment). All participants completed the Pittsburgh Sleep Quality Index (PSQI), polysomnography, the Generalized Anxiety Disorder-7 (GAD-7) scale, and the Insomnia Severity Index Scale (ISI) (see Figure 1 for detailed experimental procedures). The trial protocol was approved by the ethics committee of the First Affiliated Hospital of the Air Force Military Medical University, and the ethics committee approval number was KY20212220-F-1.

### 2.2. Patients

A total of 105 high-altitude insomnia patients, screened and qualified by physicians with professional experience in sleep medicine, were enrolled during the study period. Inclusion criteria included (1) male participants aged 18–35 years; (2) participants who met the diagnostic criteria for insomnia according to the Diagnostic and Statistical Manual of Mental Disorders, Fifth Edition [21]; (3) participants with PSQI scores > 7 and ISI scores > 15; (4) high-altitude sojourners for more than one year; (5) participants who voluntarily agreed to the survey and signed the consent form for clinical trial enrollment. Exclusion criteria included (1) participants whose insomnia was caused by organic diseases, medication, or psychiatric disorders; (2) participants who were clinically diagnosed with a severe sleep disorder or required medication; (3) those who experienced discomfort during the trial and those who failed to cooperate in completing the treatment according to the trial protocol; (4) participants with a previous history of drug or alcohol abuse; (5) participants who had taken hypnotics or psychotropic drugs two weeks prior to the start of the study.

### 2.3. Interventions

The protocol of this study is shown in Figure 1. After recording baseline information, the patients were initially assessed on clinical scales and then randomized into groups. The study was randomized and single-blinded. Patients were randomized into the tVNS, control, and CBTI groups, and only after the completion of the experiment were the participating patients unblinded.

For the stimulation site, the inner and outer surface of the tragus area of the left ear was selected for the tVNS group, and the stimulation instrument chosen was the applied vagus nerve stimulator (tVNS, Xi’an Keyue Medical Co., Ltd., Xi’an, China). The tVNS channel was connected to the ear clip, and the two ends of the clip were clamped to the medial and lateral sides of the tragus of the left ear (Figure 2). The current intensity was adjusted according to the subject’s threshold size. The electrical stimulation waveform was a single-phase rectangular pulse with a pulse width of 500 microseconds and a frequency of 25 Hz. After professional training, the patients received daily electrical stimulation as a routine treatment for 45 min each time, five consecutive times per week, with no stimulation on weekends, for a total of 20 interventions. The same stimulation site was selected for the control group and differed from the tVNS group in that stimulation was stopped after 30 s of each intervention, and the rest of the settings were kept the same as in the tVNS group.

Different from the physical intervention group, the psychotherapy in the CBTI group was carried out according to a set protocol for four weeks, with the weekly implementation of the appropriate content. Patients in the CBTI group underwent sleep therapy under the supervision of medical staff who had received in-depth training and practical experience in CBTI techniques. Over a four-week period, CBTI patients received treatment four times per week, including instruction in sleep hygiene, methods of stimulus control, cognitive recovery, relaxation techniques, and sleep restriction. A medical specialist with training in sleep medicine supervised and mentored the professionals who performed CBTI.

### 2.4. Sample Size Calculation

Based on the results of previous studies, the minimum clinically important difference in PSQI is approximately 1.14–1.75 [22]. Comparisons were made between treatment and control groups and between treatment and psychological groups, so a sample size of 27 per group would have had 90% efficacy to detect a superior effect of 1.5 for PSQI, with an alpha value of 0.025 and a beta value of 0.1. Assuming a 15% missed visit rate, a sample size of 35 participants per group was required. Therefore, the total number of participants to be randomly assigned was 105.

### 2.5. Randomisation and Blinding

Using the permutation group approach, participants were randomized sequentially, and experimental locations were randomly assigned. Before the study recruitment, a random assignment sequence was generated, and after completing the baseline assessment, all participants were divided into three groups according to the random assignment sequence. Participants in all three groups were unaware of the purpose of the study, and there was no crossover between the groups. There was no crossover between the groups of patients participating in the trial, and none were aware of the study objectives of the trial until they were unblinded.

### 2.6. Measures

The PSQI scale used in this study was created by Buysse at the Pittsburgh Medical Center in 1989 [23], in which higher scores indicate poorer sleep. It was translated into Chinese by Liu Xianchen in 1996 and used with the Chinese population; its high reliability and validity have been tested in populations of patients with insomnia [24,25]. The ISI scale, which has seven items to assess the type and symptoms of a subject’s sleep disorder, is a simple screening tool for insomnia, compiled by Morin [26]. It has been shown to have good reliability and validity in clinical insomnia patient populations [27,28]. Sleep problems such as insomnia are often accompanied by negative emotions such as anxiety, and we used the GAD-7 scale to examine these emotions in this study. The GAD-7 scale, which consists of seven questions evaluated during a recent two-week period, was used to measure the level of anxiety. The total scale score is the sum of the seven item scores [29].

All participants received a full night of dynamic PSG before and after the intervention. The polysomnograph was a 24-channel Alice6LDE polysomnograph from Philips, and the fitting of the experimental participants was completed by a medical professional, using sleepG3 software provided by experienced technicians trained in sleep scoring, sleep staging, and respiratory according to the American Sleep Society AASM criteria event determination. The main observations included total sleep time (TST), sleep efficiency (SE), sleep latency (SOL), wakefulness after sleep onset (WASO), and deep sleep (N3).

### 2.7. Statistical Analysis

Data entry and statistical analysis were performed using spss for Windows version 25.0. To describe the data frequencies, descriptive statistics were used. A Shapiro–Wilk test for the normality of continuous variables was used. In addition, the chi-square test and one-way ANOVA were used to compare the baseline data. Repeated-measures ANOVA (analysis of variance) and Bonferonni were applied to compare sleep quality and polysomnography data. A *p*-value < 0.05 was considered statistically significant.

## 3. Results

This study had 105 patients with high-altitude insomnia, of whom 35 were in the therapy group, 35 were in the control group, and 35 were in the CBTI group. There were no adverse effects in the patients who received the tVNS intervention, and 33 patients in all completed the four weeks of tVNS therapy; 2 patients discontinued the trial, and 3 patients in the control group discontinued as well. Failure to finish the whole course and taking an insomniac medicine midway through were two factors that contributed to dropouts. In the end, there were 100 patients, all of whom were male. There were no significant differences in baseline characteristics among the three groups (see Table 1).

Our results showed significant group differences in PSQI scores, ISI scores, N3, and SOL (see Table 2). The results of the repeated-measures ANOVA for PSQI scores in the tVNS group showed significant interaction between the group and number of measurements (F = 9.686, *p* < 0.001, *η^2^_p_* = 0.244), with a significant difference before and after the intervention (*p* < 0.001), indicating that the PSQI scores of insomnia patients treated with tVNS improved significantly. Similar results could be observed in SE, N3, and SOL in the tVNS group. At post-test, the PSQI scores were significantly different among the three groups (F = 15.723, *p* < 0.001, *η^2^_p_* = 0.257), with the CBTI and tVNS groups scoring significantly lower than the control group; at the fourth-week follow-up, the PSQI scores were significantly different among the three groups (F = 17.781, *p* < 0.001, *η^2^_p_* = 0.281), with the tVNS group scoring significantly lower than the control group. The tVNS group had significantly lower scores than the CBTI and control groups; at the eighth-week follow-up, the PSQI scores of the three groups were significantly different (F = 38.742, *p* < 0.001, *η^2^_p_* = 0.46), with the tVNS group having significantly lower scores than the CBTI and control groups (Figure 3). This indicates that both the tVNS technique and CBTI therapy can effectively improve insomnia in patients. Compared to CBTI therapy, both the 4-week and 8-week effects at the end of tVNS intervention were better than CBTI therapy, indicating that the long-term efficacy of the tVNS intervention technique is better than that of CBTI therapy.

The results of the repeated-measures ANOVA for the scores of the ISI scale and the GAD-7 scale showed a significant interaction between the ISI group and the number of measurements (F = 20.277, *p* < 0.001, *η^2^_p_* = 0.208) and between the GAD-7 group and the number of measurements (F = 4.414, *p* < 0.01, *η^2^_p_* = 0.094). A simple effect analysis was conducted, and it was found that in the comparison of the post-test, patients in the tVNS and CBTI groups had lower ISI and GAD-7 scores than the control group (*p* < 0.05), and this result indicated that patients in the tVNS and CBTI groups had significantly better sleep statuses and anxiety levels after the intervention than the control group; in the comparison of the fourth-week follow-up, patients in the tVNS and CBTI groups had lower ISI scores, the GAD-7 scores were lower than those of the control group (*p* < 0.05), and the ISI scores of patients in the tVNS group were lower than those of the CBTI group (*p* < 0.05), which indicated that the sleep statuses of patients in the tVNS and CBTI groups were significantly better than those of the sham-stimulated group in the fourth week after the intervention, and the improvement in the sleep of patients in the tVNS group was better than that of the CBTI group; in the comparison of scores at the eighth-week follow-up, the patients in both the tVNS and CBTI groups had lower ISI and GAD-7 scores than the control group (*p* < 0.05), and patients in the tVNS group had lower ISI scores than the CBTI group (*p* < 0.05). This result showed that patients in both the tVNS and CBTI groups improved their sleep status and anxiety significantly better than the control group at the eighth week after the intervention, and the tVNS group improved their sleep better than the CBTI group (Figure 4 and Figure 5).

A repeated-measures ANOVA on each indicator of polysomnography before and after the intervention for the three groups of subjects showed a significant main effect of the number of measurements on the N3 indicator for the three groups (F = 40.433, *p* < 0.001, *η^2^_p_* = 0.317), and post hoc comparisons showed that deep sleep duration was significantly higher on the post-test than on the pre-test level (*p* < 0.001); the interaction between the group and the number of measurements was significant (F = 3.838, *p* <0.05, *η^2^_p_* = 0.081), and a simple effects analysis revealed that at post-test, the three groups of subjects differed significantly in the N3 deep sleep period (F = 3.253, *p* < 0.05, *η^2^_p_* = 0.07), with a significantly longer time in the N3 sleep period in the tVNS group than in the control group (Figure 6). While the interaction between the group and the number of measurements on the sleep latency SOL index was significant among the three groups of subjects (F = 3.436, *p* < 0.05, *η^2^_p_* = 0.073), a simple effects analysis revealed that the sleep latency was significantly shorter in the tVNS group after the intervention compared to the pre-test level (*p* < 0.01) (Figure 7). This result indicates that the tVNS technique facilitated a more significant increase in sleep depth and in the shortening of sleep latency in subjects compared to CBTI therapy, and the intervention effect was more prominent (see Table 3).

In this randomized trial, we found that after 4 weeks of intervention, the clinical-scale PSQI, ISI, and GAD-7 scores decreased significantly in both the tVNS and CBTI groups, with significant symptom relief, indicating that both the tVNS technique and CBTI therapy were effective in improving the subjects’ insomnia status and anxiety. However, compared to CBTI therapy, the effect of the tVNS intervention was better than CBTI therapy at the end of 4 and 8 weeks. The long-term efficacy of the tVNS intervention technique was superior to that of CBTI therapy, and tVNS was effective in enhancing the depth of sleep and shortening the sleep latency of patients.

## 4. Discussion

The occurrence of altitude insomnia is closely related to neuroendocrine disorders in the human body [30]. Insomnia is often accompanied by an imbalance in the autonomic nervous system, with symptoms mostly manifesting as sympathetic hyperexcitability and vagal hypoactivity [31]. In contrast, enhanced vagal activity can facilitate the relief of insomnia symptoms. Currently, VNS has been shown to be an effective adjunctive therapy for the treatment of epilepsy, depression, insomnia, and other diseases. Many studies have shown that tVNS can act directly on the vagus nerve ear branch to regulate changes in the concentration of various neurotransmitters associated with sleep and influence brain activity to regulate sleep conditions [19]. A previous study focusing on the application of the tVNS technique to intervene in insomnia showed that patients’ sleep quality improved significantly after tVNS treatment [32]. And similarly, Marta Jackowska verified the positive effect of tVNS in improving sleep in a normal population in the community [16]. This shows the good efficacy of tVNS therapy to improve sleep.

In this study, there was no significant difference in the change in clinical scale scores between the tVNS and CBTI groups after the intervention, but the follow-up phase was better than that for CBTI therapy, indicating that the effect of tVNS in improving insomnia was comparable to CBTI therapy, but the long-term efficacy was better than that of CBTI therapy. In addition, our polysomnographic results showed that the tVNS technique facilitated a more significant increase in sleep depth and shorter sleep latency in patients compared with CBTI therapy, with a more pronounced intervention effect. This provides new evidence for the evaluation of tVNS for insomnia. This may be related to the ability of tVNS stimulation to modulate the endocrine disruption caused by the plateau environment. Other studies have shown changes in the distribution of neurotransmitters in the monoaminergic pathway and improved insomnia symptoms in rats after prolonged stimulation via VNS [33]. After comparing the changes in cortical GABA receptor density in subjects before and after VNS treatment, it was found that VNS increased the density of GABA receptors and altered the excitability of the cerebral cortex [34]. It was found that after two weeks of VNS stimulation, norepinephrine concentrations in the hippocampus and prefrontal cortex of mice were significantly increased [35]. And in another study, the metabolic rate of norepinephrine neurons was significantly increased after VNS stimulation in mice [36]. Another study reported that stimulation of the vagus nerve can affect cortical synchronization and desynchronization, increase the number of cycles of non-rapid eye movement (NREM) and Pontine–Geniculate–Occipital waves (PGO waves), and prolong the REM period [37].

Nevertheless, in this study, the subjective sleep status of patients in the tVNS group was better than that of the CBTI group during the follow-up stage. The reasons for this may be as follows: Spielman AJ proposed the 3P model of insomnia, suggesting that propensity, triggering, and retention are the three relevant factors causing insomnia, with propensity and triggering factors contributing to the onset of insomnia and retention factors maintaining it [38]. The CBTI group was set for too short a period, and the patients’ poor perceptions about sleep were not effectively improved, and the brief improvement in sleep status may have been related to the use of various relaxation techniques in CBTI therapy. In addition, the occurrence of insomnia in plateaus is mostly associated with environmental factors such as low pressure and hypoxia, and the efficacy of psychotherapy on insomnia is limited by the environment, and subjective suggestion does not counteract the effect of environmental factors on the organism to the same extent.

In addition, the anxiety levels of the patients in this study were also improved together. This may have been related to the action of tVNS on mood-related brain regions such as the amygdala and blue spot. BadranBW found, via fMRI, that tVNS caused changes in the activity of brain regions such as the hypothalamus, occipital cortex, temporal lobe, amygdala, thalamus, hippocampus, and brainstem, while the sham stimulation group in the earlobe location showed no changes in the corresponding brain regions in the non-vagal distribution [39]. In addition, Schlaepfer demonstrated that vagus nerve stimulation was effective in reducing depression in patients, and the effect was positively correlated with the duration of use [40]. FangJ found that taVNS could affect the activity of the amygdala, insula, and limbic system brain regions of the brain in emotion regulation through the vagus nerve upload pathway and alleviate the symptoms of anxious and depressed patients [41].

The study has several limitations. First, due to the limitations, only clinical symptom scale assessment was performed during the follow-up phase without polysomnography; objective sleep indicators were missing, and the results were somewhat limited. Second, the experimental subjects were all adult males, and the study results were limited in terms of gender and age representativeness. Further studies in the future will consider increasing the diversity of the sample and adding brain imaging indicators in order to explore the relevant mechanisms.

## 5. Conclusions

In conclusion, we found that in individuals with insomnia in high-altitude areas, a continuous tVNS intervention both regulated sleep and improved anxiety. Our findings have implications for a better understanding of the mechanisms of tVNS, practice for the application of tVNS in particular environments, and novel theories in the area of sleep insomnia.

## Figures and Tables

**Figure 1 brainsci-13-00985-f001:**
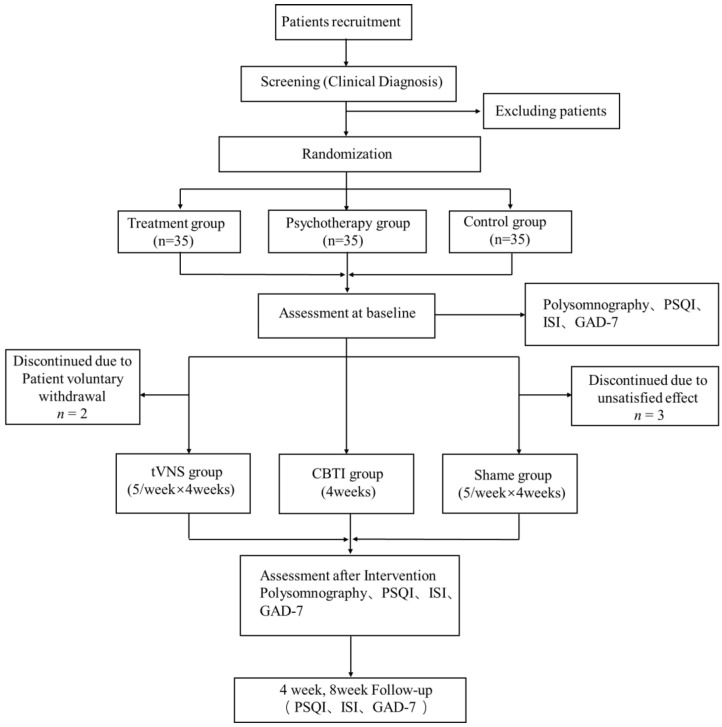
Flowchart of the research. PSQI, Pittsburgh Sleep Quality Index; ISI, Insomnia Severity Index Scale; GAD-7, Generalized Anxiety Disorder-7.

**Figure 2 brainsci-13-00985-f002:**
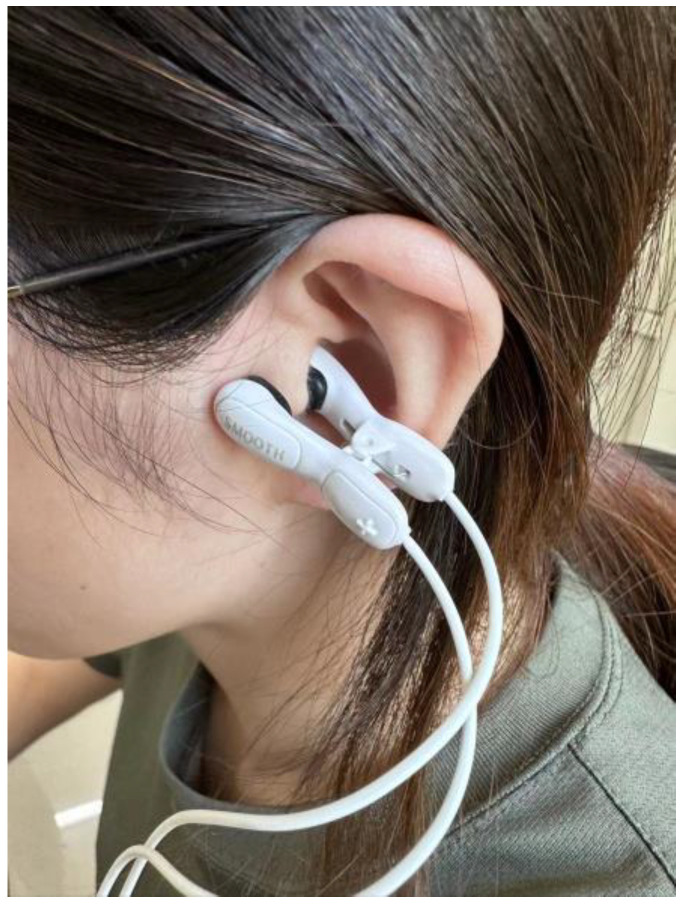
Electrode areas used for tVNS stimulation in the tVNS and control groups.

**Figure 3 brainsci-13-00985-f003:**
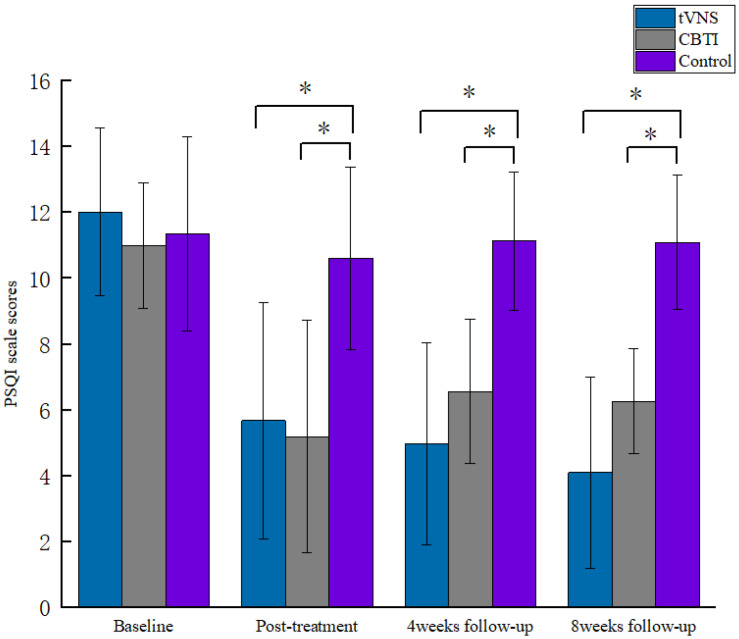
Comparison of PSQI scores by stage for each group of patients.* indicates *p* < 0.05.

**Figure 4 brainsci-13-00985-f004:**
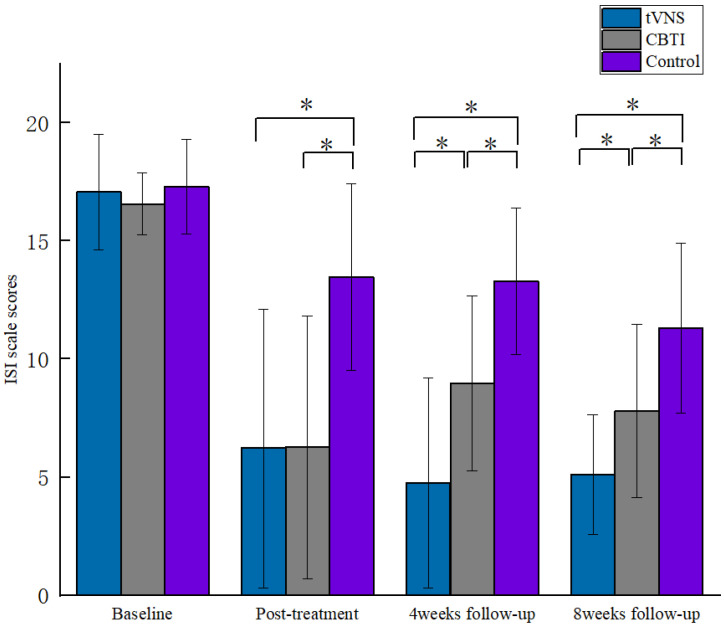
Comparison of ISI scores by stage for each group of patients. * indicates *p* < 0.05.

**Figure 5 brainsci-13-00985-f005:**
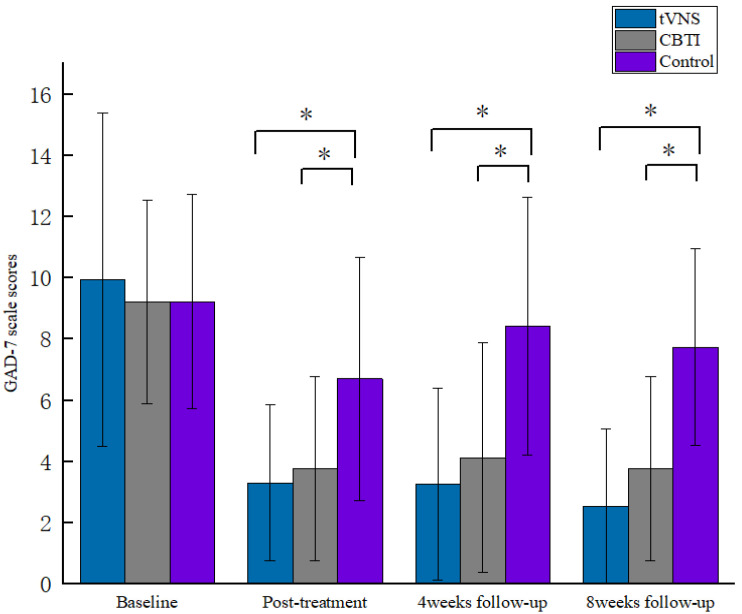
Comparison of GAD-7 scores by stage for each group of patients. * indicates *p* < 0.05.

**Figure 6 brainsci-13-00985-f006:**
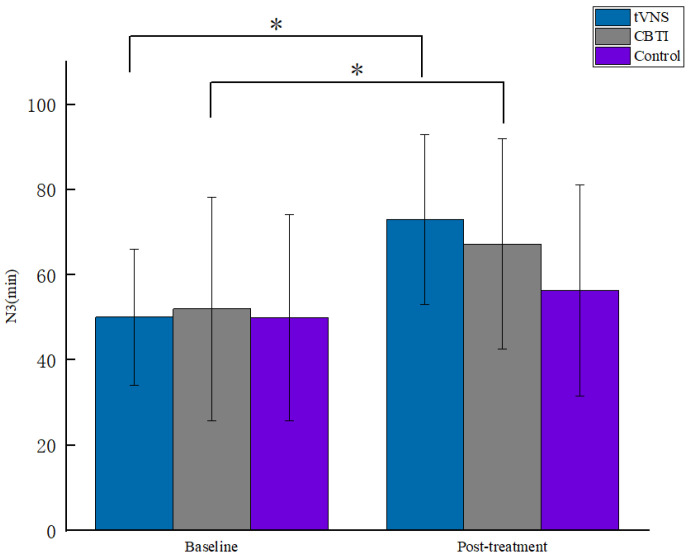
Comparison of N3 by stage for each group of patients. * indicates *p* < 0.05.

**Figure 7 brainsci-13-00985-f007:**
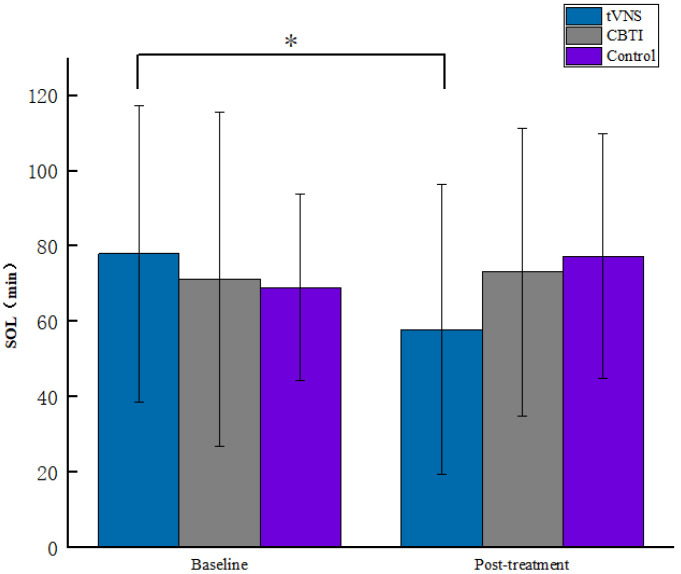
Comparison of SOL by stage for each group of patients. * indicates *p* < 0.05.

**Table 1 brainsci-13-00985-t001:** Comparisons among the three groups at baseline.

Items	tVNS Group	Control Group	CBTI Group	*F*/*χ^2^*	*p*
Sample size	33	32	35		
Age (years)	23.12 ± 2.56	23.96 ± 2.25	24.38 ± 1.95	1.797	0.185
Years of education	13.54 ± 2.01	14.17 ± 2.51	13.63 ± 2.43	0.458	0.637
Marital status				1.347	0.652
Never married	25 (75.76%)	22 (68.75%)	27 (77.14%)		
Married	5 (15.15%)	8 (25%)	6 (17.14%)		
Divorced	3 (9.09%)	2 (6.25%)	2 (5.71%)		
Pre-treatment					
PSQI	12.02 ± 2.54	11.35 ± 2.95	11.00 ± 1.91	1.577	0.212
ISI	17.07 ± 2.46	17.30 ± 1.99	16.57 ± 1.31	0.948	0.392
GAD-7	9.94 ± 5.43	9.22 ± 3.50	9.20 ± 3.32	0.305	0.738
TST (min)	403.90 ± 47.76	385.94 ± 82.88	406.68 ± 82.86	0.568	0.569
SOL (min)	77.99 ± 39.37	69.03 ± 24.72	71.23 ± 44.23	0.063	0.173
N3 (min)	50.08 ± 16.00	49.94 ± 24.17	52.00 ± 26.24	0.229	0.919
SE (%)	81.07 ± 8.59	82.97 ± 10.46	82.23 ± 11.87	0.084	0.796
WASO (min)	42.12 ± 31.61	42.40 ± 37.11	45.45 ± 53.31	0.419	0.939

Data were presented as mean ± SD. PSQI, Pittsburgh Sleep Quality Index; ISI, Insomnia Severity Index; GAD-7, Generalized Anxiety Disorder; TST, total sleep time; WASO, wake after sleep onset; SE, sleep efficiency; SOL, sleep onset latency.

**Table 2 brainsci-13-00985-t002:** Comparing tVNS vs. CBTI vs. sham on sleep and emotional status.

	Baseline	4 Weeks Post-Treatment	4-Week Follow-Up	8-Week Follow-Up	*F*, *p*
tVNS	12.02 ± 2.54	5.68 ± 3.58 ^c^	4.98 ± 3.06 ^bc^	4.10 ± 2.91 ^bc^	162.51, 0.001 *
PSQI CBTI	11.00 ± 1.91	5.20 ± 3.54 ^c^	6.57 ± 2.19 ^ac^	6.27 ± 1.60 ^ac^	20.79, 0.001 *
Control	11.35 ± 2.95	10.61 ± 2.76 ^ab^	11.13 ± 2.10 ^ab^	11.09 ± 2.04 ^ab^	71.43, 0.804
*F*	1.58	20.12	42.16	65.03	
*p*	0.212	0.001 *	0.001 *	0.001 *	
tVNS	17.07 ± 2.46	6.22 ± 5.90 ^c^	4.76 ± 4.44 ^bc^	5.10 ± 2.54 ^bc^	159.27, 0.001 *
ISI CBTI	16.57 ± 1.31	6.27 ± 5.56 ^c^	8.97 ± 3.71 ^ac^	7.80 ± 3.67 ^ac^	69.08, 0.001 *
Control	17.30 ± 1.99	13.48 ± 3.95 ^ab^	13.30 ± 3.11 ^ab^	11.30 ± 3.59 ^ab^	20.41, 0.001 *
*F*	0.95	28.51	35.87	27.96	
*p*	0.392	0.001 *	0.001 *	0.001 *	
tVNS	9.94 ± 5.43	3.31 ± 2.54 ^c^	3.26 ± 3.13 ^c^	3.31 ± 2.54 ^c^	30.99, 0.001 *
GAD-7 CBTI	9.20 ± 3.32	3.77 ± 3.00 ^c^	4.13 ± 3.76 ^c^	3.77 ± 3.00 ^c^	19.81, 0.001 *
Control	9.22 ± 3.50	6.70 ± 3.98 ^ab^	8.43 ± 4.21 ^ab^	7.74 ± 3.22 ^ab^	2.94, 0.096
*F*	0.31	9.25	14.97	18.29	
*p*	0.74	0.001 *	0.001 *	0.001 *	

* *p* ˂ 0.05. ^a^ significantly different from tVNS group. ^b^ significantly different from CBTI group. ^c^ significantly different from sham group.

**Table 3 brainsci-13-00985-t003:** Comparing tVNS vs. CBTI vs. sham on polysomnogram.

	Baseline	4 Weeks Post-Treatment	*F, p*
tVNS	403.90 ± 47.76	421.34 ± 57.41	1.07, 0.303
TST CBTI	406.66 ± 82.86	407.69 ± 69.47	0.01, 0.907
Control	385.94 ± 82.88	388.61 ± 82.61	0.004, 0.952
*F*	0.57	1.43	
*p*	0.569	0.244	
tVNS	42.11 ± 31.61	29.32 ± 24.95	1.71, 0.194
WASO CBTI	45.45 ± 53.31	45.56 ± 49.56	0.001, 0.991
Control	42.40 ± 37.11	30.17 ± 20.89	0.85, 0.359
*F*	0.06	2.08	
*p*	0.939	0.131	
tVNS	81.07 ± 8.59	86.69 ± 7.41	6.70, 0.01 *
SE CBTI	82.23 ± 11.87	83.94 ± 6.40	0.64, 0.428
Control	82.97 ± 10.46	81.88 ± 8.61	0.14, 0.711
*F*	0.23	2.90	
*p*	0.796	0.06	
tVNS	50.08 ± 16.00	72.97 ± 19.88 ^c^	40.87, 0.001 *
N3 CBTI	52.00 ± 26.24	67.24 ± 24.69	18.62, 0.001 *
Control	49.95 ± 24.17	56.29 ± 24.81 ^a^	1.70, 0.195
*F*	0.08		
*p*	0.919		
tVNS	77.99 ± 39.37	57.93 ± 38.35	5.23, 0.025 *
SOL CBTI	71.23 ± 44.23	73.17 ± 38.19	0.05, 0.823
Control	69.03 ± 24.72	77.37 ± 32.33	0.491, 0.486
*F*	0.42	2.24	
*p*	0.659	0.112	

* *p* ˂ 0.05. ^a^ significantly different from tVNS group. ^c^ significantly different from sham group.

## Data Availability

The corresponding author can provide the study data upon reasonable request.

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
