# Peer review of "Transcutaneous Vagus Nerve Stimulation for Insomnia in People Living in Places or Cities with High Altitudes: A Randomized Controlled Trial"

_brainsci, 2023, doi:10.3390/brainsci13070985_

Round 1
Reviewer 1 Report
The introduction lacks depth. The link with the obstructive sleep apnea hypopnea syndrome and altitude as well as with vagal nerve stimulation is not enough described. The authors also reported studies about high altitude insomnia for people ''new to the plateau'' but the study is about people living on the plateau for more than one year. The flow through the introduction but also through the paper and makes it very hard to follow.
Some inclusion criteria are questionable, why only men, at least justify? Was it because it was a military population? If yes, it needs to be clearer that the studied population was military. I don't understand why people recruited but with who did not completed the protocol were considered in the exclusion criteria. If difficulties happened, they should be reported, as well as side effects. On the electrical current, a more specific localisation than the left ear is needed. 5 times a week, were the days consecutive? Why did you chose to do 30 sec stimulation over ear lobe stimulation for sham? Was the CBTI individual therapy or group therapy? Tables need to have more complete definition at the bottom. Some abbreviations are missing.
The association between melatonin and the vagal nerve stimulation is a weak hypothesis, which means that even if it could make sense, it is not based on literature or requires more justification and references. In the discussion, the phrasing makes it unclear if the information has been established, hypothesized or extrapolated.
I would like to see more limitations regarding the design.
Choice of words to reassess: sojourn, plateau without adjective on the altitude, etc.
Introduction and discussion need to be clearer and more organized.
There is a subtitle error.
Author Response
Dear Editor,
We are pleased to answer the questions of the reviewers’ and the manuscript has also been extensively revised according to the comments (resubmitted online).
Kind regards,
Zhang Liang
Point#1:The introduction lacks depth. The link with the obstructive sleep apnea hypopnea syndrome and altitude as well as with vagal nerve stimulation is not enough described.
Response 1:The introduction part has been reworked.
Point#2:The authors also reported studies about high altitude insomnia for people ''new to the plateau'' but the study is about people living on the plateau for more than one year. The flow through the introduction but also through the paper and makes it very hard to follow.
Response 2:People who have just arrived at the plateau will have insomnia mainly because the body cannot adapt to the plateau environment, and this point is mentioned mainly to highlight that the plateau environment is the main cause of insomnia. This section has been revised.
Point#3:Some inclusion criteria are questionable, why only men, at least justify? Was it because it was a military population?
Response 3:The group of choice is the military, but due to policy and regulatory restrictions need to be declassified.
point#4:I don't understand why people recruited but with who did not completed the protocol were considered in the exclusion criteria. If difficulties happened, they should be reported, as well as side effects. On the electrical current, a more specific localisation than the left ear is needed. 5 times a week, were the days consecutive? Why did you chose to do 30 sec stimulation over ear lobe stimulation for sham? Was the CBTI individual therapy or group therapy? Tables need to have more complete definition at the bottom. Some abbreviations are missing.
Response 4:The above sections have been revised in the reworked draft.
Point#5:The association between melatonin and the vagal nerve stimulation is a weak hypothesis, which means that even if it could make sense, it is not based on literature or requires more justification and references. In the discussion, the phrasing makes it unclear if the information has been established, hypothesized or extrapolated.
Response5:The Discussion section has been reworked.

Reviewer 2 Report
Thank you for the opportunity to review this manuscript, which presents an interesting idea, however, the study has many flaws and problems that need to be better explained by the authors. My suggestions are outlined below.
Title
- The title is confusing the term “insomnia at high altitudes” makes the sentences confusing. I suggest changing it to “insomnia in people living in places or cities with high altitudes”.
- I suggest changing the term “clinical” to “controlled” in the title, as the authors used in the abstract to standardize throughout the study.
Abstract
- Remove numbers from the abstract (1), (2) ...
- Where is the purpose of the study?
- The authors must standardize the terms used in the study, in some sentences they use the term “High altitude”, including in the title, and in other places they use “Highland” or “setting of the plateau”... the authors must standardize the terms throughout the entire study.
- I believe that the authors should expose the results of the comparison between the tVNS versus CBTI groups, as informing that the therapy is more effective than not being exposed to anything is a somewhat predictable and expected result.
Introduction
- I suggest authors make the introduction into five paragraphs, currently the introduction has very large paragraphs. This change will make the introduction more harmonic in terms of length for readers.
- I believe the authors should find a better word to connect the two sentences below, or to start the second sentence.. “for CBTI cannot be met in China under the country's current medical system. The negative effects of temazepam ...”. The way it is, the change of sentence that ended talking about public medical services in China and adverse drug events was very abrupt.
- I suggest that the authors review the internationally used term about: Repetitive Transcranial Magnetic Stimulation (tVNS) and update it throughout the study.
- The authors use the abbreviation TMS without explaining what it means first, initially inserting the name in full and then using the abbreviation.
- Where is the purpose of the study at the end of the introduction? Authors need to insert.
Methods
- The methods are described in a way in which the information is duplicated, that is, the authors speak twice or more of each step, they talk about the volunteers, the randomization process... and then they talk about each step again. Authors must rewrite the methods explaining how the study occurred temporally. Reporting the steps several times makes reading confusing and tiring for readers, I suggest rewriting the methods taking as reference the time in which each of the steps occurred.
- How was the randomization performed? What methods did the authors use?
- Was the clinical trial protocol registered on any international clinical trial registration platform? Like clinical trials, for example? If not, authors should include this methodological flaw as a serious limitation of their study.
- I strongly suggest that the authors insert photos of the data collection, which demonstrate how and where the electrodes were inserted in the patients in the tVNS group, for a better understanding of the study and also, because, as the authors mentioned, there are few studies on the subject . Thus, this better understanding of how the stages of this study were carried out can increase the chances of citations of this study by researchers in the area in future investigations on the subject.
- Why did the CBTI group have a different number of sessions and time compared to the tVNS group? When the clinical trial aims to compare therapies, similar interventions must be offered to the groups in terms of session time, weekly sessions and total intervention time. This difference between interventions may be a bias in this study.
- Were the PSQI and the other instruments used validated for the Chinese population? If so, it must be described and referenced.
- I suggest once again a better description of the intervention with photos of the equipment used and how it is applied to the volunteers for a better understanding of the readers
- Why did the authors use only ANOVA, no 2-by-2 comparison?
Results
- The authors must correct the number of volunteers who participated in the study throughout the study. In the summary the information is that 105 volunteers participated in the study, when we mention the word participated in the study it means that they started and ended the study. However, the total seems to have been 100 volunteers, this is also confusing in figure 1, which does not show the number of final volunteers after losses, I suggest adding it to figure 1 as well.
- The way in which the statistical analysis took place is also confusing. The use of ANOVA is correct, it tells me whether or not there are differences between the three groups, however as it stands I don't know where those differences are. The authors should make the comparison between two groups to demonstrate where these statistical differences are.
- As it stands I cannot understand whether or not there are differences between the groups through direct comparisons, for example: tVNS versus CBTI; tVNS versus control; CBTI versus control.
Discussion
- This study should answer the following question for readers: Which therapy is most effective in reducing my insomnia or improving my tVNS or CBTI sleep? As the results are presented, the authors are not able to answer this question for readers, however, they have the data to answer. I suggest improving the explanation of the results in order to answer this study question.
- The discussion is a little lacking in scientific arguments. What scientific model do the authors believe could support their findings?
- The authors use the acronym NE without explaining beforehand what it is about.
- The authors do not explain how the use of tVNS physiologically or scientifically can improve sleep quality or reduce insomnia. The authors also cite several assumptions and only cite reference 41, this sentence was confusing.
- Were there any adverse events from the intervention? Authors need to mention, even if there weren't any.
- The authors mention that the study is single-blind, but they never mention the blinding of any stage of the study. If there was no blinding, the authors need to add as one more limitation of the study.
Conclusion
- In this section the authors should only mention the conclusion of the study, clinical implications or suggestions for future investigations on the subject can be mentioned at the end of the discussion.
The text presents many terms in which the authors used different words for their translation. I suggest standardizing the terms used throughout the manuscript and improving verbal and nominal agreement throughout the article.
Author Response
Dear Editor,
We are pleased to answer the questions of the reviewers’ and the manuscript has also been extensively revised according to the comments (resubmitted online).
Kind regards,
Zhang Liang
Point#1:The title is confusing the term “insomnia at high altitudes” makes the sentences confusing. I suggest changing it to “insomnia in people living in places or cities with high altitudes” I suggest changing the term “clinical” to “controlled” in the title, as the authors used in the abstract to standardize throughout the study.
Response 1:The above part has been reworked.
Point#2:Remove numbers from the abstract (1), (2) .. Where is the purpose of the study. The authors must standardize the terms used in the study, in some sentences they use the term “High altitude”, including in the title, and in other places they use “Highland” or “setting of the plateau”... the authors must standardize the terms throughout the entire study.I believe that the authors should expose the results of the comparison between the tVNS versus CBTI groups, as informing that the therapy is more effective than not being exposed to anything is a somewhat predictable and expected result.
Response 2:The above part has been reworked.
Point#3: I suggest authors make the introduction into five paragraphs, currently the introduction has very large paragraphs. This change will make the introduction more harmonic in terms of length for readers. I believe the authors should find a better word to connect the two sentences below, or to start the second sentence.. “for CBTI cannot be met in China under the country's current medical system. The negative effects of temazepam ...”. The way it is, the change of sentence that ended talking about public medical services in China and adverse drug events was very abrupt.I suggest that the authors review the internationally used term about: Repetitive Transcranial Magnetic Stimulation (tVNS) and update it throughout the study.The authors use the abbreviation TMS without explaining what it means first, initially inserting the name in full and then using the abbreviation.Where is the purpose of the study at the end of the introduction? Authors need to insert.
Response 3:All the above have been modified
point#4:The methods are described in a way in which the information is duplicated, that is, the authors speak twice or more of each step, they talk about the volunteers, the randomization process... and then they talk about each step again. Authors must rewrite the methods explaining how the study occurred temporally. Reporting the steps several times makes reading confusing and tiring for readers, I suggest rewriting the methods taking as reference the time in which each of the steps occurred.How was the randomization performed? What methods did the authors use? Was the clinical trial protocol registered on any international clinical trial registration platform? Like clinical trials, for example? If not, authors should include this methodological flaw as a serious limitation of their study. I strongly suggest that the authors insert photos of the data collection, which demonstrate how and where the electrodes were inserted in the patients in the tVNS group, for a better understanding of the study and also, because, as the authors mentioned, there are few studies on the subject . Thus, this better understanding of how the stages of this study were carried out can increase the chances of citations of this study by researchers in the area in future investigations on the subject.Why did the CBTI group have a different number of sessions and time compared to the tVNS group? When the clinical trial aims to compare therapies, similar interventions must be offered to the groups in terms of session time, weekly sessions and total intervention time. This difference between interventions may be a bias in this study.Were the PSQI and the other instruments used validated for the Chinese population? If so, it must be described and referenced. I suggest once again a better description of the intervention with photos of the equipment used and how it is applied to the volunteers for a better understanding of the readersWhy did the authors use only ANOVA, no 2-by-2 comparison?
Response 4:All the above have been modified

Round 2
Reviewer 1 Report
In the Figure 1, I would specify the follow-up timepoint because for 2 months seems to general.
According to Figure 2, how is it possible to distinguish the stimulation site between cymba conchae and the tragus?
In the discussion, when Badran's results are discussed, the sham used was different from the one in this study. I think it should be specify that the ear lobe was the sham to avoid confusion.
I still have question about the population studied in regard of their occupation. If participants are military, when not specify it. It could influence generalization and their life experience could also be different from civilian participants.
Why did you use the first name in the text when referring to authors?
Correct:
Line 86: group. patients
Line 104: period.Inclusion
In the list of inclusion/exclusion critera, please be consistent with the ponctuation
In figure 1: unsatisfied effecte
Lines 133, 374, 381 spacing
Lin 343 : similarly Marta jackowska
I think that this version is improved a lot, well done!
Author Response
Dear Editor,
We are pleased to answer the questions of the reviewers’ and the manuscript has also been extensively revised according to the comments (resubmitted online).
Kind regards,
Zhang Liang
Point#1:In the Figure 1, I would specify the follow-up timepoint because for 2 months seems to general.
Response 1:Modifications have been made to Figure 1
Point#2:According to Figure 2, how is it possible to distinguish the stimulation site between cymba conchae and the tragus?
Response 2:I have misunderstood the translation, the actual stimulation site of the experiment is the inner and outer surface of tragus.
Point#3:In the discussion, when Badran's results are discussed, the sham used was different from the one in this study. I think it should be specify that the ear lobe was the sham to avoid confusion.
Response 3:This sentence has been modified
point#4:I still have question about the population studied in regard of their occupation. If participants are military, when not specify it. It could influence generalization and their life experience could also be different from civilian participants.
Response 4:The subjects were from the plateau military group, and the trial was designed to solve the insomnia problem of plateau military personnel, but due to the policy document restrictions, the information of the subjects should be treated confidentially.
Point#5:Correct:Line 86: group. Patients; Line 104: period. Inclusion; In the list of inclusion/exclusion critera, please be consistent with the ponctuation ;In figure 1: unsatisfied effecte; Lines 133, 374, 381 spacing; Lin 343 : similarly Marta jackowska
Response5:The above sections have been revised.

Reviewer 2 Report
Congratulations to the authors, they did a good job. All my suggestions/requests were answered. I believe that as it stands, the article achieved an improvement in its quality and is now ready to be accepted for publication.
Author Response
Dear Editor,
We are pleased to answer the questions of the reviewers’ and the manuscript has also been extensively revised according to the comments (resubmitted online). For some details in the article, we have made some changes, please check
Kind regards,
Zhang Liang
Point#1:In the Figure 1
Response 1:Modifications have been made to Figure 1
Point#2: Figure 2
Response 2:I have misunderstood the translation, the actual stimulation site of the experiment is the inner and outer surface of tragus.
Point#3:In the discussion, when Badran's results are discussed, the sham used was different from the one in this study. I think it should be specify that the ear lobe was the sham to avoid confusion.
Response 3:This sentence has been modified
